# Phosphate-Solubilizing Bacteria: Advances in Their Physiology, Molecular Mechanisms and Microbial Community Effects

**DOI:** 10.3390/microorganisms11122904

**Published:** 2023-12-01

**Authors:** Lin Pan, Baiyan Cai

**Affiliations:** 1Engineering Research Center of Agricultural Microbiology Technology, Ministry of Education & Heilongjiang Provincial Key Laboratory of Ecological Restoration and Resource Utilization for Cold Region, Key Laboratory of Molecular Biology, College of Heilongjiang Province, School of Life Sciences, Heilongjiang University, Harbin 150080, China; panlin0610@126.com; 2Hebei Key Laboratory of Agroecological Safety, Hebei University of Environmental Engineering, Qinhuangdao 066102, China

**Keywords:** phosphorus, phosphorus-solubilizing bacteria, mechanisms of phosphorus solubilization, microbial community effects, molecular mechanism

## Abstract

Phosphorus is an essential nutrient for all life on earth and has a major impact on plant growth and crop yield. The forms of phosphorus that can be directly absorbed and utilized by plants are mainly HPO_4_^2−^ and H_2_PO_4_^−^, which are known as usable phosphorus. At present, the total phosphorus content of soils worldwide is 400–1000 mg/kg, of which only 1.00–2.50% is plant-available, which seriously affects the growth of plants and the development of agriculture, resulting in a high level of total phosphorus in soils and a scarcity of available phosphorus. Traditional methods of applying phosphorus fertilizer cannot address phosphorus deficiency problems; they harm the environment and the ore material is a nonrenewable natural resource. Therefore, it is imperative to find alternative environmentally compatible and economically viable strategies to address phosphorus scarcity. Phosphorus-solubilizing bacteria (PSB) can convert insoluble phosphorus in the soil into usable phosphorus that can be directly absorbed by plants, thus improving the uptake and utilization of phosphorus by plants. However, there is no clear and systematic report on the mechanism of action of PSB. Therefore, this paper summarizes the discovery process, species, and distribution of PSB, focusing on the physiological mechanisms outlining the processes of acidolysis, enzymolysis, chelation and complexation reactions of PSB. The related genes regulating PSB acidolysis and enzymatic action as well as genes related to phosphate transport and the molecular direction mechanism of its pathway are examined. The effects of PSB on the structure and abundance of microbial communities in soil are also described, illustrating the mechanism of how PSB interact with microorganisms in soil and indirectly increase the amount of available phosphorus in soil. And three perspectives are considered in further exploring the PSB mechanism in utilizing a synergistic multi-omics approach, exploring PSB-related regulatory genes in different phosphorus levels and investigating the application of PSB as a microbial fungicide. This paper aims to provide theoretical support for improving the utilization of soil insoluble phosphorus and providing optimal management of elemental phosphorus in the future.

## 1. Introduction

Phosphorus is an essential plant macronutrient second only to nitrogen and is directly involved in nucleic acid synthesis, cell division and the growth of new tissues and is also necessary for a variety of cellular processes such as photosynthesis, carbohydrate metabolism, energy production, redox homeostasis and signaling in plants [1,2,3]. Phosphorus is also essential for plant growth and development, with levels reaching 0.2% of the dry weight of plants [4]. However, arable land worldwide contains more total phosphorus and less available phosphorus, and the concentration of plant-available phosphorus seldom exceeds 10 μM, which seriously affects plant growth and agricultural development [5].

Soil organic phosphorus accounts for 30–65%, while the remaining 35–70% is in inorganic forms. Organic phosphorus generally exists in an inert form in the soil and is fixed to form insoluble phosphorus forms, which are difficult for plants to directly absorb and use, and inorganic phosphorus easily reacts with ions in the soil, such as Fe^3+^, Al^3+^ and Ca^2+^, generating insoluble phosphate; thus, the long-term application of phosphorus fertilizer overwhelmingly results in phosphate that is difficult for plants to directly absorb and cannot completely address phosphorus deficiencies in soils [6,7]. The overapplication of phosphate fertilizers can also lead to environmental problems such as negative impacts on the food chain, eutrophication, soil nutrient imbalances and destruction of the soil microbial environment. Global phosphate reserves are limited, and depletion will probably occur in 50–100 years (approximately 2059–2109) [8]. However, these data could be changed and updated as new phosphate ores have been recently discovered in Scandinavian countries such as Norway. Due to the associated deleterious effects of the application of phosphorus fertilizers and the decline in global phosphorus stocks, efforts must be made to find alternative environmentally compatible and economically viable strategies to improve phosphorus levels in low-phosphorus or phosphorus-deficient soils for plant growth [1].

The discovery of PSB provides a new way to solve the problem of effective phosphorus deficiency in soils; PSB play a key role in the soil phosphorus cycle by mineralizing organic phosphorus through the secretion of acid and hydrolyzing inorganic phosphorus minerals through enzyme activity, thus solubilizing insoluble phosphorus and increasing the amount of available phosphorus in soils [9,10]. Several studies in recent years have confirmed that PSB can convert the insoluble phosphorus forms present in soils into available phosphorus that can be directly absorbed and utilized by plants through different mechanisms. Examples include secretions of enzymes (phytase and phosphatase), acids (organic, inorganic) and chelation (siderophores, extracellular polysaccharides) to produce dissolved phosphate [10,11]. In addition to their own ability to alter phosphorus speciation, PSB can act synergistically with phosphate fertilizers, and their joint application has been shown to improve phosphorus utilization [12]. PSB are also gradually becoming raw materials for biofertilizers and inoculants; the effect of their application in agricultural fields is also noticeable and can significantly increase the productivity of agronomic crops in agroecological niches [13,14]. Currently, research on PSB mechanisms is limited in scope and lacks comprehensive studies from different perspectives. Due to the interference of PSB species and external factors, PSB metabolites have different levels of capacity, and many functional genes have different effects at different levels, while the ecological environment in soil is so complex that both the exogenous addition of PSB and fertilization will trigger changes in microbial communities, which indirectly change the phosphorus content in soil. Based on these findings, this paper summarizes the phosphorus solubilization mechanism of PSB in soil and the problems encountered from different perspectives, including physiological, molecular and microbial community effects, to provide a theoretical basis for reducing the use of chemical fertilizers and improving the utilization of soil phosphorus in the future (Figure 1).

## 2. Overview of PSB

A relationship between microorganisms and soil phosphorus was first noticed at the beginning of the 20th century, when organisms capable of mineralizing phosphate, called phosphorus-solubilizing microorganisms, were discovered for the first time in nature [15]. In 1948, Gerrestsen reported the effect of microorganisms on phosphorus uptake by plants and found that the dry weight of plants grown in normal soil increased by 72% to 188% compared with plants grown in sterilized soil and phosphorus uptake increased by 79% to 342%. In the same year, the first experiments on phosphorus-solubilizing microorganisms for plant growth were carried out, and it was found that phosphorus-solubilizing microorganisms could promote plant growth. Since the greatest proportion of microorganisms in the soil is bacteria, scientists began to study PSB [16]. Initial progress was made in the 1950s in the study of the mechanism of phosphorus solubilization, and it was noted that PSB could secrete different organic acids and their dephosphorylating ability was related to the pH of the surrounding environment [17].

### Distribution and Species of PSB

PSB in soil can convert insoluble phosphorus forms into plant-available phosphorus; these bacteria are mostly Gram-negative (Table 1) [18]. There are many types of PSB, which can be classified into two main groups according to the different substrates they act on: organic PSB, which can mineralize organic phosphorus, and inorganic PSB, which can convert insoluble inorganic phosphorus into soluble organic phosphorus; many PSB are also able to dissolve inorganic phosphorus and mineralize organic phosphorus at the same time [19,20]. Phosphorus-solubilizing microorganisms are composed of phosphorus-solubilizing bacteria, phosphorus-solubilizing actinomycetes and phosphorus-solubilizing fungi, among which PSB are widely distributed, accounting for 1–50% of phosphorus-solubilizing microorganisms in the soil, phosphorus-solubilizing actinomycetes for 15–45% and phosphorus-solubilizing fungi for only 0.1% to 0.5%. [21,22]. Djuuna et al. found the highest proportion of PSB among the phosphorus-solubilizing microorganisms screened in the soil, with a total population ranging from 25 × 103 to 550 × 103 CFU g^−1^ [23]. The distribution of PSB is also strongly influenced by many factors. For example, soil properties, habitat types and rhizosphere effects all contribute to differences in the distribution of PSB, and these distributional differences can directly affect PSB species and abundance [17] (Figure 2). Currently, high-throughput sequencing technology is used to determine the abundance of PSB communities, to understand the state of their distribution and to analyze the diversity of PSB communities in the soil and their composition. The effects of different factors on the distribution of PSB are also different. Currently, *Bacillus* spp., *Burkholderia* spp. and *Pseudomonas* spp. show high distribution and abundance in all soil types. Numerous studies have identified *Bacillus* spp., *Pseudocystis* spp. and *Burkholderia* spp. in different types of soils (tea gardens, saline soil, soils with heavy metals, forest soil) and in crop rhizosphere soils, where they have a high relative abundance in the bacterial community as well as a strong phosphorus-solubilizing capacity [24,25,26,27,28,29,30,31]. Currently, other PSB such as *Enterobacter*, *Salmonella*, *Flavobacterium*, *Micrococcus*, *Thiobacillus*, *Azotobacter*, *Burkholderia*, *Enterobacteriaceae*, *Serratia marcescens* and *Baeyerlingia*, are also present in the soil (Table 1) [32,33]. In addition, many different PSB were also isolated using genome sequencing (high-throughput sequencing) and raw plate isolation at different distribution sites. Table 2 and Table 3 describe the distribution of PSB by type of crop and by location, respectively.

The species and capacity of PSB isolated from different types of habitats vary. Tang et al. screened eight species of PSB in forest soils, of which more than one-third were from the genus *Burkholderia* and all of which had good phosphate-dissolving capacity [31]. Most of the PSB isolated from oil tea plantations by Liu et al. were *Bacillus*, with *Bacillus ayabatensis* (NC285) having the greatest ability to solubilize inorganic phosphorus [24]. Gao et al. screened the rhizosphere soils of sand dunes for PSB, which contained a high proportion of *Bacteroidetes* spp.; *Bacteroidetes* spp. had a higher proportion in soils with better nutrient supplies [34]. Xie et al. screened fruit tree rhizosphere soils in karst rock desertification (KRD) and non-karst rock desertification (NKRD) areas for PSB and found that the number of PSB screened in the KRD area was much higher than that in the NKRD area; *Pseudomonas* spp. were present in the soils of both the KRD and NKRD areas, whereas *Fusobacterium* spp. and *Bakkholderia* spp. appeared only in some of the soil samples from the KRD area [35]. Therefore, there is a need for targeted strain screening based on differences in geography and plant species.

Soil properties also affect the distribution of PSB, with different soil properties causing differences in the distribution of PSB with different functions. Krystal et al. isolated 44 strains of PSB from calcareous soil, of which 15 strains were identified as *Bacillus flexus*, *Beijerinckia fluminensis*, *Enterobacter ludwigii*, *Enterobacter* sp., and *Pantoea cypripedii*. Among these, *Enterobacter ludwigii* was the best PSB; most of the PSB isolated from soils with a high cadmium content belonged to *Bacillus* and *Enterobacteriaceae*, and the isolated PSB were highly tolerant of cadmium [25]. Rahul et al. isolated two PSB, *Pseudomonas carboxylans* and *Pseudomonas malodorans*, from soils heavily contaminated with heavy metals, and these bacteria were extremely tolerant of lead [26]. Zhang et al. found that bacteria isolated from saline soils belonged to *Bacillus*, such as *Bacillus amyloliquefaciens*, and these PSB were better for the amelioration of saline soils [36]. As a result of these distributional characteristics, PSB can be used to remediate specific soils.

The distribution of PSB also showed an obvious rhizosphere effect. The number and activity of PSB in rhizosphere soils were significantly higher than those in non-rhizosphere soils; this effect played a key role in the construction of PSB communities in rhizosphere soils and influenced the transformation of substances, nutrient cycling and energy flow in soils [37]. The rhizosphere effects of PSB have also been demonstrated in several experiments. Li et al. demonstrated in their experiments that the number of PSB isolated from rhizosphere soils was significantly greater than the number of PSB isolated from non-rhizosphere soils [38]. Zheng et al. analyzed the gene abundance of phosphatase and the *phoC* and *phoD* functional genes of PSB in maize rhizosphere and nonrhizosphere areas; *phoC* and *phoD* encode acid phosphomonolipase and alkaline phosphomonolipase, respectively [39]. The researchers found that phosphatase activity and the gene copy numbers of *phoC* and *phoD* were higher in rhizosphere soils than in non-rhizosphere soils, which indicated that rhizosphere soils possessed higher PSB activity [40]. The strains screened from different plant roots varied; for example, *Bacillus* spp. were the dominant strains in rice, *Fusobacterium* spp., *Bacillus* spp., *Enterobacteriaceae* and *Ochrobacterium* spp. were the dominant strains in wheat, and most of the PSB screened from tea roots were strains of *Burkholderia* spp. [41,42,43]. Similar results were found by Ponmurugan and Gopi, who determined the population densities of PSB in the rhizospheres of peanut, duckweed, sorghum and maize, with results of 14.9 × 105, 8.6 × 105, 7.3 × 105 and 7.7 × 105 Cfu/g, respectively, indicating that PSB had the highest population densities in the peanut rhizosphere, while PSB had relatively small population densities in the rhizosphere of the remaining three crops. The population densities were relatively low in the other three crops [23,44].

At present, people have a certain understanding of the distribution characteristics and patterns of PSB, but this understanding is still incomplete. PSB, as the largest proportion of phosphorus-solubilizing microorganisms, are distributed all over the world. In some off-the-beaten-track or extreme environments of unexplored areas, PSB and other characteristics of the distribution of the bacteria still need to be explored.

**Table 1 microorganisms-11-02904-t001:** Physiological and biochemical testing of different PSB.

PSB	Gram Stain	Glucose Hydrolysis	Starch Hydrolysis	Gelatin Liquefaction	Citrate Utilization	Hydrogen Sulfide Generation	V-P	Methyl Red	Reference
*Pantoea roadsii*	-	+		+	+	-	-	+	[45]
*Pseudomonas donghuensis*	-	+		+	+	-	-	+	[45]
*Ochrobactrum pseudogrignonense*	-	+		-	-	-	-	+	[45]
*Pseudomonas moraviensis*	-	+	+	+	+		+		[46]
*Bacillus safensis*	+	+	+	+	+		+		[46]
*Falsibacillus pallidus*	-	+	-	-	-		-		[46]
*Pseudomonas* sp.	-		-	-				-	[47]
*Acinetobacter calcoaceticus*	-		-	-				-	[47]
*Antagonistic Bacillus*	+		+	+	+	+	+	-	[48]

Note: In the table, “+” is a positive reaction result and “-” is a negative reaction result.

**Table 2 microorganisms-11-02904-t002:** Distribution of PSB in rhizosphere soils of different crop types.

Crop Species	Distribution	PSB	Impact on Crop Performance	Reference
Cereals	Maize rhizosphere soil	*Pseudomonas fluorescens*, *Pseudomonas poae*, *Bacillus subtilis*	Increase in dry weight	[49]
	Wheat rhizosphere soil	*Bacillus safensis*, *Falsibacillus pallidus*	Root length, root surface area, root volume and number of root tips increased significantly	[46]
	Maize rhizosphere soil	*S. marcensens*, *P. brenneri*	Substantial increase in maize production	[50]
Pulses	Legumes in Fes-Meknes region rhizosphere soil	*PSB WJEF38*	Agronomic traits of peas and broad beans were improved	[51]
	*Phaseolus vulgaris* L. rhizosphere soil	*Pseudomonas kribbensis*	Biomass increased dramatically	[13]
	Peanut rhizosphere soil	*Bacillus amyloliquefaciens*	Significant increase in aboveground dry and fresh weights	[48]
Horticultural crops	Walnut, feijoa, jujube, apple rhizosphere soil	*Pantoea gavini, Acinetobacter* sp.	Increase in root length	[52]
	*Chenopodium quinoa* rhizosphere soil	*Licheniformis, Enterobacter, Asburiae,*	Increase in fresh weight and root length	[53]
	Tomato rhizosphere soil	*Acinetobacter, Stenotrophomonas maltophilia*	Tomato plant height and leaf area were increased	[54]
Cash crops (economics)	Cotton rhizosphere soil	*Bacillus halotolerans*	Increased cotton yield	[55]
	Tobacco rhizosphere soil	*Burkholderia cenocepacia*	Significant increase in plant height	[56]

**Table 3 microorganisms-11-02904-t003:** Distribution of PSB in different types of sites and screening tools.

Source	Source Location	PSB General Category	Screening Method	Reference
Soil	Arid land	*Pseudomonas azotoformans*, *Acinetobacter baumannii*, *Bacillus paramycoides*	Flatbed screening	[57]
	Cinnamomum camphora soil	*Bacteroidetes*, *Proteobacteria*, *Chloroflexi*, and *Gemmatimonadetes*	High-throughput sequencing	[45]
	Saline soil	*Bacillus amyloliquefaciens*	Flatbed screening	[36]
	Brazilian cerrado soil	*Pseudomonas aeruginosa*, *Bacillus cereus*	Flatbed screening	[58]
Rhizosphere	Rhizosphere of Taxus chinensis var. mairei	*Pseudomonas fluorescens*, *Bacillus cereus*, *Sinorhizobium meliloti*, *Bacillus licheniformis*	Flatbed screening	[59]
	Rice rhizosphere	*Pseudomonas aeruginosa*, *Bacillus subtilis strain*	Flatbed screening	[41]
	Blueberry plant rhizosphere	*Buttiauxella* sp.	High-throughput sequencing	[60]
	Characteristics of rhizosphere	*Ascomycetes*, *Acidobacteria*,	High-throughput sequencing	[61]
Sediment	Reservoir sediment	*Micromonospora* sp., *Aminobacter* sp.	High-throughput sequencing	[62]
	Surface sediment in the Changjiang or Yangtze River estuary	*Firmicutes, Proteobacteria*, *Actinobacteria*	Flatbed screening	[63]
	Lake Taihu sediment	*Burkholderia* sp.	Flatbed screening	[64]
Plant parts	Roots, stems and leaves of moso bamboo	*Alkaloid-producing bacilli of the genus Bacillus*, *Enterobacter* spp., *Bacillus* spp.	Flatbed screening	[65]
	Stems of Oryza officinalis	*Acinetobacter, Cutibacterium*, *Dechloromonas*	Flatbed screening	[66]
	Corn roots	*Burkholderia* spp.	Flatbed screening	[67]

## 3. Mechanisms of Phosphorus Solubilization by PSB

Phosphorus solubilization mechanisms are mainly concerned with important components of the soil phosphorus cycle (dissolution–precipitation, mineralization–fixation and adsorption–desorption), which, in addition to being related to bacterial species, are also regulated by phospholysis-related genes. Most of the phosphate-solubilizing bacteria can mineralize or hydrolyze the insoluble phosphate in the soil by secreting acids and enzymes and also chelate or complex with metal cations (Ca^2+^, Fe^2+^, Al^3+^) in the soil to release phosphate ions. A small number of phosphate-solubilizing bacteria can indirectly change the pH of the environment by releasing gas molecules (CO_2_ through respiration and hydrogen sulfide from PSB) to achieve the effect of solubilizing insoluble phosphate. PSB can also increase the phosphorus content of soils by altering the microbial community in the soil. PSB in the soil change the diversity of the microbial community and the associated enzymes increase, thus changing the phosphorus content of the soil and enhancing crop growth and yield [68] (Figure 3). Due to the complexity and diversity of phosphate solubilization mechanisms, this paper focuses on three systematic aspects: the physiological mechanisms of PSB, molecular mechanisms and indirect mechanisms shaped by changes in the structure and abundance of microbial communities in the soil (Figure 4).

### 3.1. Physiological Mechanisms

#### 3.1.1. Solubilizing Action of Acids

Acidolysis involves the action of both organic and inorganic acids; most PSB secrete organic acids and very few secrete inorganic acids. PSB bacteria can secrete low-molecular-weight organic acids such as malic acid, succinic acid, fumaric acid, acetic acid, tartaric acid, malonic acid, glutamic acid, propionic acid, butyric acid, lactic acid, 2-ketogluconic acid, saccharinic acid and oxalic acid during growth [69,70]. These low-molecular-weight organic acids can chelate under low pH conditions via hydroxyl and carboxyl groups with metal ions in the soil (Fe^3+^, Al^3+^, Ca^2+^) (Figure 5) [71]. Their Ca^2+^ chelation ability is most significant, and they compete with phosphates for phosphorus adsorption sites in the soil, enhancing the soil uptake of phosphate and increasing the solubilizing capacity of inorganic phosphorus, resulting in increased solubility and availability of mineral phosphates [72,73]. Solubilization efficiency is also related to the nature of the organic acids; tricarboxylic and dicarboxylic acids are usually more effective than mono- and aromatic acids, and aliphatic acids are more effective than phenolic, citric and fumaric acids in solubilizing phosphates [74]. However, different types of organic acids are produced by different PSB, and different organic acids have very different adsorption capacities for phosphate (Table 4) [75,76]. A few PSB also secrete inorganic acids (e.g., hydrochloric, sulfuric, nitric and carbonic acids), which lower the soil pH and dissolve inorganic phosphorus but are not as effective as organic acids at the same pH [33].

PSB secretes organic acids that interact with plant roots [77]. This rhizosphere effect, which increases the rhizosphere’s bacterial diversity and the relative abundance of dominant bacteria in the rhizosphere zone, indirectly alters the available phosphorus content of the soil [78,79]. The acidic environment created by organic acids is more suitable for the enrichment of PSB, but it is also somewhat targeted. For example, malic and citric acids can both attract *Pseudomonas fluorescens* WCS365, which can colonize tomato roots [80]. Watermelon roots can secrete citric and malic acids, which induce plants to promote the colonization of roots by a rhizosphere bacterial strain, *Bacillus polymyxa* SQR-21 [81]. Exploring the production of organic acids in the plant root system–PSB–plant root system relationship is of great value in refining the mechanism of phosphate solubilization.

Although most studies have concluded that the secretion of organic acids is one of the main mechanisms of PSB, a few still have different views on this. Zhao et al. found that the organic acid content of the medium did not correlate with the phosphorus-solubilizing capacity of PSB in their experiments [82]. A similar conclusion was reached by Qin et al., who found that the application of PSB to over-acidified soils increased the pH and that there was no positive correlation between organic acids and the ability to dissolve phosphorus [83]. These results showed that the secretion of organic acids did not necessarily have a strong correlation with phosphate-solubilizing ability, and the important mechanism between organic acids and phosphate-solubilizing bacteria was not fully explored and still needs to be further investigated.

**Table 4 microorganisms-11-02904-t004:** Species of organic acids secreted by different PSB.

Phylum	PSB Species	PSB Source	Secreted Organic Acids	References
Proteobacteria	*Enterobacter aerogenes*	Mangrove rhizosphere soil	Lactic, succinic, isovaleric, isobutyric and acetic acids	[84]
	*Pantoea dispersa*	Corn rhizosphere soil	Citric, malic, succinic and acetic acids	[85]
	*Pantoea* sp.	Nectarine rhizosphere soil	Oxalic, formic, acetic and citric acids	[86]
Firmicutes	*Bacillus*	Rice paddy soil	Gluco-oxalic acid, citric acid, tartaric acid, succinic acid, formic acid and acetic acid	[87]
	*Bacillus safensis*	Turmeric rhizosphere soil	Gluconic acid, alpha-ketogluconic acid, succinic acid, oxalic acid and tartaric acid	[69]
	*Bacillus siamensis*	Wheat rhizosphere soil	Glycolic acid	[88]
	*Bacillus amyloliquefaciens*	Saline soil	Lactic acid, maleic acid and oxalic acid	[36]
Actinobacteria	*Tsukamurellatrosinosolvens*	Tea tree rhizosphere soil	Lactic acid, maleic acid and oxalic acid	[89]

#### 3.1.2. Mineralization Action of Enzymes

The mineralization of organophosphorus by enzymes secreted by PSB is one of the main dephosphorylation mechanisms (Table 5). The main enzymes found to have a dephosphorylating effect are phosphatases, phytases and C-P-cleaving enzymes. It has also been well demonstrated in many experiments that PSB increase enzyme activity in the soil, resulting in an increase in the available phosphorus content of the soil, but different hydrolytic enzymes are involved in the mineralization of organic phosphorus in different ways [88,90]. Phosphatase, also known as phosphomonolipase, is mainly encoded by *olp*A and is classified into three types: acidic, alkaline and neutral phosphatases. ACP is more effective in mineralizing organic phosphorus in acidic soils with pH values less than 7. Alkaline phosphatases (ALPs) mainly catalyze the hydrolysis of phospholipids (i.e., phosphoglucose-6 and ATP) and release inorganic phosphorus in soils with pH values higher than 7. In contrast, neutral phosphatases, unlike acid and alkaline phosphatases, do not have a very pronounced effect on phosphorus mineralization and hydrolysis. [2,87,91]. Phytase is another type of phosphatase (encoded by *appA*) that mainly mineralizes organic phosphorus in phytate. As the ester bond in phytate is quite stable, phosphatases are not able to completely hydrolyze it, and it must be converted by phytase into a form that can be broken down by phosphatases, finally releasing inorganic phosphorus [92]. C-P lyase (encoded by *phnJ*) is involved in the mineralization of organophosphorus, as are several of the enzymes mentioned above, but differs in that the hydrolysis of organophosphorus by C-P lyase ultimately releases phosphorus in its free form [2].

#### 3.1.3. Chelation and Complexation

Chelation and complexation are two important mechanisms of phosphorus solubilization that are based on the principle of the functional group binding to metal cations in the soil, releasing phosphate for the purpose of phosphorus solubilization. Currently, chelation and complexation are produced mainly through siderophores produced by phosphorus-solubilizing bacteria, extracellular polysaccharides and the decomposition of plant and animal residues. Siderophores are small molecules with low molecular weights, and under low iron stress, phosphate-dissolving bacteria produce siderophores that chelate Fe^3+^, Al^3+^ and Ca^2+^ in the soil, releasing phosphate ions for phosphate dissolution [71,100,101]. The formed metal-iron carrier complexes can bind to iron carrier receptor proteins on cell membranes to enter cells, increasing iron uptake by plants and promoting plant growth [25]. Extracellular polysaccharides secreted by PSB are high-molecular-weight sugar polymers attached to bacterial surfaces that have special anionic functional groups (phosphate, carboxyl and succinate groups) and abundant -OH and -COOH acid groups on the surface that can complex with cations of heavy metal elements in the soil [102]. During the degradation of plant and animal residues by PSB, humic acid-like substances are formed that can also chelate Fe^3+^, Al^3+^ and Ca^2+^ in the soil, releasing phosphate and increasing the available phosphorus content in the soil [17].

### 3.2. Molecular Mechanisms

In addition to the physiological mechanism, the molecular mechanism of PSB is also important, but at present the complete molecular mechanism of the PSB pathway is not clear in the research; it is mainly studied through the genome sequencing of PSB’s metabolism of acid and enzyme-related functional genes, such as *GDH*, *pqq*, *ALP*, *ACP,* and so on. Thus, the mechanism of PSB metabolism is gradually formed through the combination of different functional genes and the metabolites they regulate.

#### 3.2.1. Functional Genes Related to the Regulation of Acidolysis

The production and secretion of organic acids are the main mechanisms of phosphorus solubilization by PSB, but the molecular mechanism of acidolysis has not been studied in depth, and there are only a few genes that can play a role in regulating phosphate solubilization (Table 6) such as *gadγ* (Gluconate 2-dehydrogenase), *gdh* (Glucose dehydrogenase), *gcd* (Glucose dehydrogenase) and *ylil* (Aldose sugar dehydrogenase) [103,104,105]. In addition to the proven *gcd* genes, pyruvate dehydrogenase (*poxB*), l-lactate dehydrogenase (*ldh*), glyoxylate reductase (*gyaR*) and glyoxylate/hydroxypyruvate/6-ketoglucuronate reductase (*ghrB*) are several genes associated with organic acid metabolism that may be relevant to phosphate solubilization by PSB [104]. However, among the many genes, gluconic acid-related genes have been more intensively studied, and many Gram-negative bacteria use periplasmic glucose oxidation, mediated by pyrroloquinoline quinone (*PQQ*) and glucose dehydrogenase (*GDH*), to produce gluconic acid that encodes *GCD*; pyrroloquinoline quinone acts as an oxidative cofactor in glucose dehydrogenase and is mediated by the *pqq* manipulator genes (A, B, C, D, E, F) [2,105,106]. Most *PQQ* genes are present in the genus Pseudomonas [107]. Among the pqq A-E genes, *pqqA*, *pqqC*, *pqqD* and *pqqE* are prerequisite genes for phosphate catabolism by PSB [108]. Moreover, *pqqA* and *pqqC* are the two key genes; the gene *pqqA* consists of 22 amino acids, the peptides of glutamic and tyrosine that provide carbon and nitrogen to the *PQQ* biological system, and the gene *pqqC* that encodes the pyrroloquinoline synthase C (*PQQC*) gene, which catalyzes the conversion of 3a-(2-amino-2-carboxyethyl)-4,5-dioxo-4,5,6,7,8,9-hexahydroquinoline-7,9-dicarboxylic acid to pyrroloquinoline quinone (*PQQ*) [109]. It has been shown that the *pqqC* gene is highest in many PSB, such as *Burkholderia* (8.98–16.69%), *Azotobacter* (5.27–11.05%), *Pseudomonas aeruginosa* (3.31–9.28%) and *Mycobacterium* (1.25–9.8%), and the *pqqC* gene has also been identified to track the inorganic phosphate-solubilizing ability marker genes of microorganisms [99]. Joshi et al. found 16 bacterial isolates with 82 bp *pqqC* gene amplification positivity and 6 bacterial isolates with 72 bp *pqqA* gene amplification positivity. Bacterial isolates exhibiting *pqqA* gene amplicons were also positive for the *pqqC* gene [110]. The results of these experiments demonstrated that there are two key genes in the *PQQ* biosynthetic pathway. The production of many organic acids, such as citric acid, α-ketoglutaric acid, succinic acid and malic acid, requires the TCA cycle, and genes related to the dephosphorylation mechanism may be relevantly linked to the TCA cycle. Ding et al. showed that the inoculation of PSB in insoluble phosphorus form into a medium resulted in a significant enhancement of TCA-related genes such as citrate synthase *(CS*), isocitrate dehydrogenase (*IDH*), α-ketoglutarate dehydrogenase (*OGDH*), succinate dehydrogenase (*SDH*) and fumarate hydratase *(FH*) compared to the CK (not inoculated with phosphorus-solubilizing bacteria) [111]. Studies on the TCA cycle and the acidolysis mechanism of PSB are scarce, and these topics still need to be explored in depth.

#### 3.2.2. Functional Genes Related to the Regulation of Enzymolysis

Many of the carrier genes in PSB encode different enzymes (Table 7), such as alkaline phosphatases (*phoD* and *phoA*), acid phosphatases (*phoC*), phytase (*appA*), C-P cleavage enzyme (*phn*), extracellular polyphosphatase (*ppx*) and polyphosphate kinase (*ppk*), with alkaline phosphatases (*phoD* and *phoA*) being the most studied [39,112]. Alkaline phosphatase production in PSB is encoded by one of three homologous gene families (*phoA*, *phoD*, *phoX*) as part of the pho regulator. Macrogenomics studies have also shown that 32% of sequenced prokaryotic genomes contain at least one *phoA*, *phoD* or p*hoX* gene [113]. Under inorganic phosphorus deficiency conditions, especially, *phoA* and *phoD* are the most common genes, with *phoD* also being the most strongly linked to alkaline phosphatase [20]. Wan et al. also showed that inoculation with PSB increased the abundance of genes related to organic and inorganic phosphorus cycling, including *phoD*, and that the *phoD* copy number was positively correlated with soil alkaline phosphatase activity [11,114]. There are also numerous genes that can participate in the regulation of the phosphorus deprivation response (*phoU*, *phoR* and *phoB*), which are closely linked to genes involved in phosphorus uptake and transport (*pst*) and control the expression of genes encoding alkaline phosphatases, especially under low phosphorus conditions [112]. In their experiments, Dai et al. found that under low phosphorus conditions, the release of phosphatase could be enhanced by regulating a gene involved in the regulation of the phosphorus starvation response (*phoR*) to mineralize soil organophosphorus. In contrast, a consistently higher abundance of the *phoR* gene was observed only in low-phosphorus soils compared to high-phosphorus soils [112]. Although alkaline phosphatase is more likely to be secreted by soil PSB, acid phosphatase is still secreted in small amounts. Additionally, bacterial genera carrying pho genes may contain acid phosphatase genes, such as *Pseudomonas*, *Stemonella* and *Lysobacillus* [109,114].

Phytase is mainly encoded by the *bpp* gene, which is more widely distributed than other phytate degradation-related genes (*ptp* and *hap*), and there is also a strong positive correlation between *bpp* gene abundance and the number of phytate-degrading bacteria [94]. The *bpp* gene has a six-bladed propeller-folded structure and contains a cleavage, an affinity phosphate binding site and six calcium binding sites, three each for enzyme stability and catalysis. To date, only a small number of *bpp* in PSB have been isolated and investigated, e.g., *Bacillus subtilis* and *Bacillus licheniformis* [5]. In addition, the *phn* gene in C-P lyase and the genes for extracellular polyphosphatase (*ppx*) and polyphosphate kinase (*ppk*) biosynthesis can all solubilize inorganic phosphate (*polyP*) [115].

The study of phospholysis-related genes is of transgenerational significance, revealing the metabolic mechanism of phosphate-solubilizing microorganisms in solubilizing insoluble phosphorus at the genetic level, and we have witnessed the excavation of many phospholysis genes. Nevertheless, information about some unknown phosphate solubilization mechanisms and their related genes is still limited, and some pathways are still waiting to be discovered. Most of the studies at the present stage mainly focus on the genes coding for the synthesis of organic acids, such as gluconic acid and phosphatases. Studies on individual genes are restricted to the level of functional validation. In-depth research on the metabolic mechanisms of the upstream and downstream processes is still relatively limited, and the interactions and links between the related genes are not yet clear [60].

**Table 6 microorganisms-11-02904-t006:** Genes related to the acidolytic action of different PSB.

PSB Species	PSB Source	Related Genes	Functions	Reference
*Pseudomonas putida*	Laboratory storage	*gcd*	Encode glucose dehydrogenase, which promotes the	[116]
			solubilization of inorganic P	
*Pseudomonas* sp.	Wheat rhizosphere soil	*gcd*	Encode glucose dehydrogenase, which promotes the solubilization of inorganic P	[105]
*Acinetobacter*	Soils in rocky desertification areas	*gcd*	Mediate the production of gluconic acid	[35]
*Acinetobacter pittii gp-1*	Laboratory storage	*gcd*	Promotion of the solubilization of inorganic and organic phosphorus	[94]
*Ochrobactrum haematophilum*	Sweet potato rhizosphere soil	*CS*, *ACO*, *ODGH*, *SFD*, *FH*, *MDA*	Tricarboxylic acid cycle-related genes	[111]
*Ochrobactrum haematophilum*	Sweet potato rhizosphere soil	*POX*, *LDH*	Acetic acid and lactic acid regulatory genes	[111]
*Acinetobacter* spp., *Pseudomonas* spp.	Rice rhizosphere soil	*pqq*C*, pqq*E	Regulation of gluconic acid production	[107]

**Table 7 microorganisms-11-02904-t007:** Genes related to the enzymolysis action of different PSB.

PSB Species	PSB Source	Related Genes	Functions	Reference
*Ochrobactrum* sp	Wheat rhizosphere soil	*pho*	Promotion of the solubilization of inorganic and organic phosphorus	[117]
*Pantoea agglomerans*	Wheat rhizosphere soil	*phy*	Participate in the dissolution of phytic acid	[117]
*Acinetobacter pittii gp-1*	Laboratory storage	*phoD*, *bpp,*	Promotion of the solubilization of inorganic and organic phosphorus	[94]
*Arthrobacter*	Corn rhizosphere soil	*Ppx*, *ppk*	Promotes the synthesis of exonuclease polyphosphatase and polyphosphate kinase	[94]
*aryabhattai*	Laboratory storage	*Phn*, *pho,*	Promotion of phosphorus metabolic pathway activity	[95]
*Pseudomonas*	Corn rhizosphere soil	*bpp*	Phytase-encoding genes	[118]
*Pantoea brenneri*	Soil samples of the Republic of Tatarstan	phnK	C-P lyase regulatory genes	[119]

### 3.3. Mechanisms of Microbial Community Effects

Current studies on PSB mechanisms mainly focus on bacteria. In addition to bacterial mechanisms, changes in soil nutrients also have a certain impact on the bacterial phosphorus-solubilizing effect. On the other hand, the relationship between PSB and indigenous microorganisms in the soil can also directly affect the structure and function of soil microorganisms, changing phosphorus transformation in the soil [120].

#### 3.3.1. Effects of Soil Nutrient Changes on the Abundance of PSB Communities

The phosphorus-solubilizing capacity of PSB is related to the abundance of their community, which is closely related to nutrient changes in the soil. Exogenously applied fertilizers can improve soil nutrients and influence the abundance of phosphate-dissolving bacterial communities through pH and C:N:P stoichiometry [121]. Changes in the composition and activity of PSB communities are driven by the C:P and N:P ratios in the soil [116]. Several experiments have confirmed that the sustained, exogenous addition of inorganic phosphorus, biochar and organic carbon increased pH and that the abundance of PSB communities increased with pH, which was positively correlated with biomass, phosphorus content and phosphorus uptake [122,123,124,125]. By contrast, the effect of nitrogen fertilizer application on community abundance is not clear; moderate nitrogen fertilizer application can improve the soil nitrogen–phosphorus balance, improve the growth conditions of PSB, and increase the abundance of PSB communities [126]. However, high concentrations of nitrogen fertilizer can inhibit the growth and proliferation of PSB because nitrogen fertilizer application increases the nitrogen content of the soil, changes the nutrient environment and relative nitrogen–phosphorus ratios of soil microorganisms and decreases soil pH. PSB may be inhibited, which may lead to a decrease in their abundance [51,127,128]. Conversely, appropriate amounts of nitrogen reduction may be positive for the abundance of PSB communities [129]. The interventions of C, N and P all resulted in different changes in pH and corresponding changes in the structure of the PSB community. It has been shown that soil pH affects the abundance of PSB communities. In acidic environments, which contain more protons, it is easy to release insoluble phosphorus, so the abundance of PSB communities does not change significantly under normal circumstances, while in alkaline environments, because a large amount of residual phosphorus is immobilized by the binding of metal ions, the abundance of PSB communities rises significantly to assist in the release of insoluble phosphorus; therefore, the higher the pH value of the soil, the greater the abundance of the PSB community [130].

#### 3.3.2. Effects of PSB on Soil Microcosm Systems

Current agricultural practices negatively impact soil microbial composition and biodiversity and are unsustainable [131]. In contrast, the effect of PSB on soil microorganisms is more positive. The introduction of PSB into soils can have an effect on both the diversity and abundance of the indigenous microbial community; the abundance and diversity of indigenous microorganisms may be enhanced, but some experiments have shown that the addition of PSB resulted in a significant decrease in the diversity of the indigenous bacterial community, an increase in the abundance of the inoculated PSB and an enhancement of the soil phosphorus content [132,133]. Bacterial communities with higher abundance are also positively correlated with the growth of some substances associated with the phospholysis mechanism (phosphatase, phytase, etc.) [134]. Thus, the contribution of the bacterial community to phosphorus solubilization can be indirectly improved by the interaction of PSB and indigenous microorganisms, suggesting that there is also a link between elevated soil phosphorus levels and changes in microecosystems, which indicates a potential mechanism for phosphorus enhancement. Therefore, further attention should be given to the effects of exogenous microbial inoculants, including PSB applied to calcareous soils, on the diversity and composition of soil bacterial communities.

In summary, the mechanism of phosphorus solubilization by PSB is complex and varied, and the physiological mechanism includes acidolysis, enzymatic action for mineralization and hydrolysis of insoluble phosphate and chelation and complexation for binding of cations in soil. The molecular direction covers the genes and pathways involved in the regulation of acidolysis and enzymolysis and with phosphate transport. In addition to the above internal regulatory mechanisms, PSB also have external regulatory mechanisms that alter the structure and abundance of microbial communities in the soil to indirectly achieve the dissolution of insoluble phosphorus. [134].

### 3.4. PSB Regulates Plant Root Transporter Proteins

PSB in the soil allow the conversion of insoluble phosphorus in the soil into phosphorus that can be directly absorbed and utilized by the plant, resulting in an increase in phosphorus content in the plant. This is not only related to the genes of PSB themselves but also to the interaction between PSB and the root system. PSB can regulate an inorganic phosphate transporter protein in plant roots to promote phosphorus uptake in plants in the presence of phosphorus deficiency. PSB can regulate an inorganic phosphate transporter protein in plant roots to promote phosphorus uptake in plants in the presence of phosphorus deficiency [20]. Srivastava et al. showed that *Pseudomonas* putida regulates *PHT1* and *PHT2* and relieves phosphorus deficiency under low phosphorus stress [135]. Similar conclusions were reached in the trial by Saia et al. where *PHT1* gene expression was increased in wheat due to inoculation with Bacillus, promoting phosphorus uptake in wheat [136]. Moreover, the elevated expression of inorganic phosphate transporter proteins due to PSB regulation will cause plants to develop smaller root systems with more lateral branches and root hairs, which stimulate phosphorus transporter proteins that produce phosphatases to solubilize phosphorus that is not available to plants [137]. Thus, PSB have significant implications for improving plant uptake of available phosphorus by inducing inorganic phosphate transport proteins in the plant root system.

## 4. Problems and Future Outlook

### 4.1. Multi-omic Synergy in the In-Depth Mining of Functional Genes of PSB

The use of multi-omic synergy can increase the rate of research progress on the mechanism of dephosphorylation. Macrogenomic technology can be used to analyze genes that are significantly up- or downregulated by PSB in different environments, GO (gene ontology) can be used to functionally annotate them and QS (group sensing) analysis can reveal the regulatory mechanism of PSB group effects. Joint transcriptome and metabolome analyses to uncover differential genes can be used to quickly determine the core regulatory networks and key candidate genes, clarify the functions of transcription factors and ultimately screen for functional genes in mechanistic studies. Ding et al. used multi-omic synergy to investigate the mechanism of PSB, and the macrogenomic results identified the genes of the main enzymes and their coenzymes involved in the synthesis of gluconic acid, 2-ketogluconic acid and indole acetic acid (IAA), and the transcriptomics analyses of differential genes in PSB with GO enrichment and KEGG enrichment revealed the relevant gene expression levels and their regulatory proteins related to the synthesis pathway of organic acids. Transcriptomics analysis of differential gene expression levels and regulatory proteins in PSB revealed the types and amounts of organic acids secreted by PSB, and metabolomics results also showed the types and amounts of organic acids secreted by phosphorus-solubilizing bacteria [111]. Currently, there are some similarities and differences in the related genes and pathways obtained by whole genome sequencing and enrichment analyses of different PSB. However, this can also prove that the study of the whole genome sequence of PSB may be an important and new research direction to gain insights into its mechanism (Table 8). Therefore, multi-omics crossover will also open up new avenues for the study of phospholysis mechanisms.

### 4.2. PSB Regulatory Genes in Soils with Different Phosphorus Levels

As the molecular mechanisms of PSB continue to be explored, the number of PSB regulatory genes has gradually increased, but soil phosphorus content and the types of refractory phosphorus will make the abundance and types of related genes different. Sun et al. succeeded in experimentally predicting five related genes that can solubilize inorganic phosphorus and mineralize organic phosphorus: *ppa* (encoding inorganic pyrophosphatase), *phoD* (encoding alkaline phosphatase D), *phnW* (encoding 2-aminoethylphosphonate pyruvate aminotransferase), *phnX* (encoding phosphorylglycolide hydrolase), *phnA* (encoding phosphonoacetic acid hydrolase) and *phnA* (encoding phosphonoacetic acid hydrolase). These five genes were significantly enriched in phosphorus-deficient soils, and their total abundance was higher than that in soils with normal phosphorus levels. However, the relative abundance of two other genes, *gcd* (encoding pentaprotein glucose dehydrogenase) and *appA* (encoding 4-phosphatase/acid phosphatase), was reduced in phosphorus-deficient soils compared to normal-phosphorus soils [134]. Lv et al. found the highest abundance of inorganic phosphorus-solubilizing genes (*gcd*, *pqqC* and *ppa*), the second-highest abundance of phosphorus deprivation-regulated genes (*phoR*) and the lowest abundance of genes involved in the mineralization of organic phosphorus (*appA* and 3-phytase) in the soil [144]. Therefore, continuous summarization and understanding of the types and abundance of genes regulated by PSB under different phosphorus levels are also among the components of the continuous deepening of understanding of the molecular mechanism of PSB.

### 4.3. Use of PSB for Making Microbial Preparations in Agriculture

PSB have the function of improving soil fertility, remediating land pollution, reducing heavy metals on land and restoring the ecological environment. Microbial agents made from dephosphorylating bacteria have great potential. For example, microbial preparations using *Pantoea* sp. and *Pseudomonas* sp. as formulas have been proven to be effective in promoting the growth of lucerne, and *Bacillus licheniformis* and *Pseudomonas aeruginosa*, alone or in combination, can also be effective in promoting the growth of Brassica campestri alone or in combination [134]. At present, about 80 countries in the world are carrying out the promotion of microbial preparations, but there are still some unsolved problems. For example, the backward production process leads to insufficient inoculum quantity, which is not enough to achieve the effect of a stable yield increase. There are fewer new varieties of microbial preparations that are applicable to a single environment [145]. In addition, in terms of the safety of dephosphorylating bacteria, sometimes there is a certain threat of dephosphorylating microorganisms promoting conditions for pathogenic bacteria, such as the strong pathogen *Klebsiella pneumoniae*, so the application should be screened to prevent the negative impacts of the spread of the application and to make safety the first priority [146]. In addition to the problems of microbial agents in production applications, there are also concerns about how much higher crop yields and economic benefits can be achieved by applying microbial agents to the soil. The emergence of machine learning models has been applied to assess the effectiveness of phosphorus-solubilizing microorganisms in agricultural remediation, and it is believed that the application of machine learning models to predict the effectiveness of microbial agents in agricultural production also has a promising future [147].

## Figures and Tables

**Figure 1 microorganisms-11-02904-f001:**
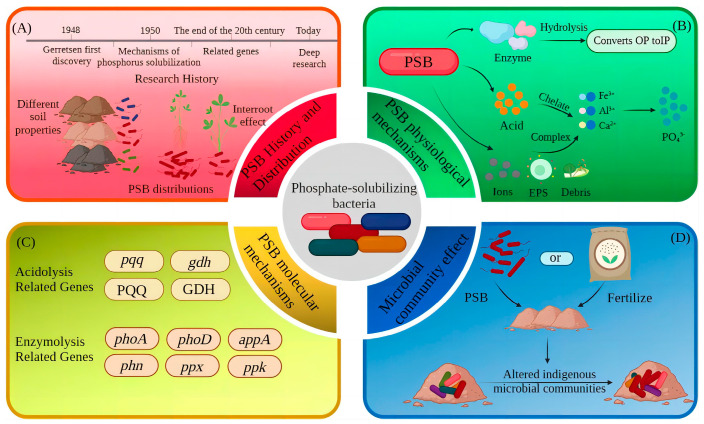
Research process of PSB phosphorus solubilization mechanism. Note: PSB = phosphorus-solubilizing bacteria; PQQ = pyrroloquinoline quinone; GDH = glucose dehydrogenase; PSB have been explored for their phosphorus-solubilizing mechanism from the time they were discovered and have been progressively refined from physiological to molecular to microbial community effects. (**A**–**D**) describe the developmental history and species distribution of PSBs, the physiological mechanisms of PSBs, the molecular mechanisms of PSBs, and the effects of PSBs on microbial communities, respectively.

**Figure 2 microorganisms-11-02904-f002:**
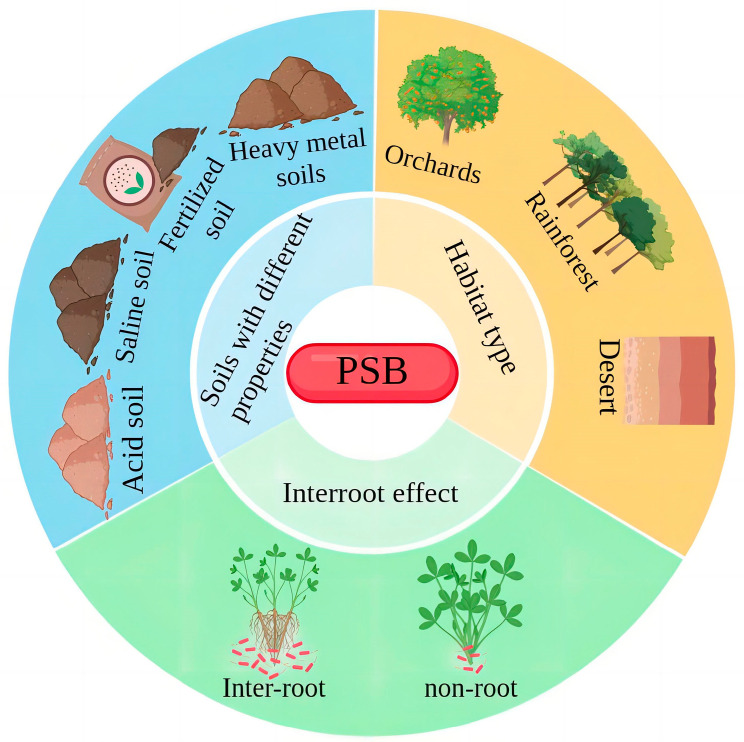
PSB distribution characteristics. Note: PSB = Phosphorus-solubilizing bacteria; the distribution of PSB is mainly influenced by three aspects: the nature of the soil, habitat type and rhizosphere effects.

**Figure 3 microorganisms-11-02904-f003:**
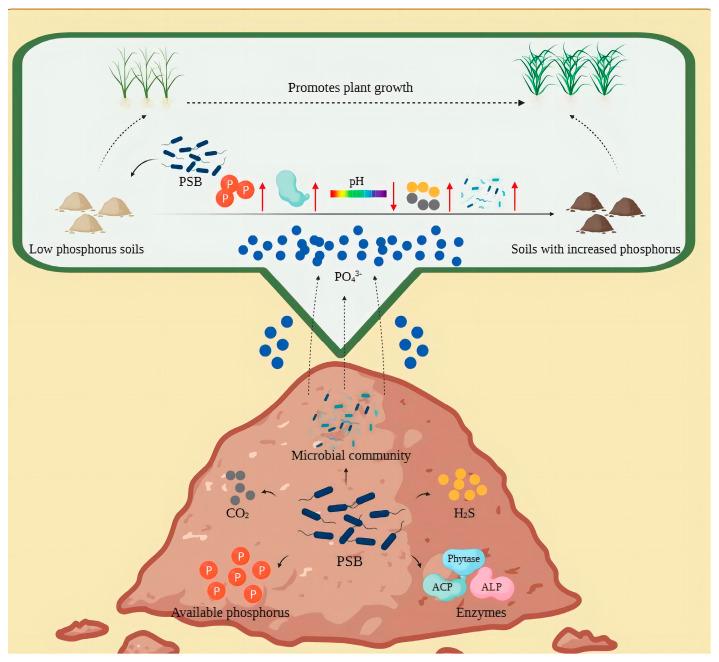
Substances involved in phosphorus solubilization by PSB. Note: PSB = phosphorus-solubilizing bacteria; CO_2_ = carbon dioxide; H_2_S = hydrogen sulfide. PSB are mainly affected by organic acids, related enzymes, gas molecules such as CO_2_ and H_2_S and microbial communities in the process of phosphorus solubilization, which generally leads to a decrease in pH and achieves a certain effect of phosphorus solubilization. Enhanced soil phosphorus levels increase crop growth and fields.

**Figure 4 microorganisms-11-02904-f004:**
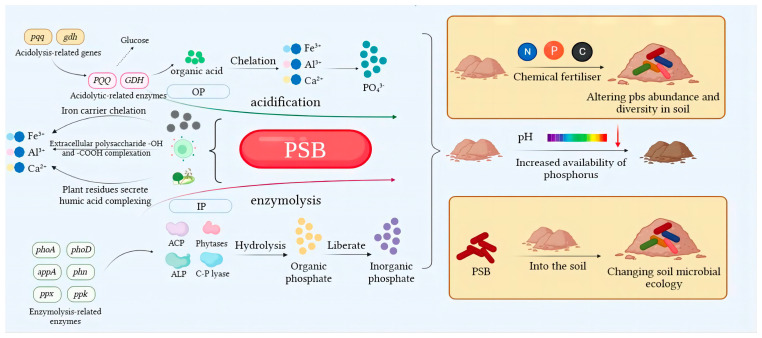
Mechanism of phosphorus solubilization by PSB. Note: PSB = phosphorus-solubilizing bacteria; OP = organic phosphate; IP = inorganic phosphate; ACP = acid phosphatase; ALP = alkaline phosphatase; PQQ = pyrroloquinoline quinone; GDH = glucose dehydrogenase; PSB dissolve/mineralize insoluble phosphorus forms in soil through different mechanisms. Acidolysis occurs in organic phosphorus soils, in which the pqq and gdh genes promote the secretion of organic acids under the regulation of PQQ and GDH, respectively, thereby chelating metal ions (Ca^2+^, Al^3+^, Fe^3+^) in the soil to release PO_4_^3−^. Enzymatic degradation occurs in inorganic phosphorus soils, where phoA, appA, ppx, phoD, phn, ppk and other related genes promote the synthesis of related enzymes to hydrolyze organic phosphorus in the soil, thereby releasing inorganic phosphorus. In addition, PSB will cause iron ions, extracellular polysaccharides, plant and animal residues and metal ions to undergo chelating and complexation reactions to reduce soil pH and increase the available phosphorus in the soil. External fertilizers and access to PSB can change the abundance of the soil microbial community, which can increase the available phosphorus in the soil.

**Figure 5 microorganisms-11-02904-f005:**
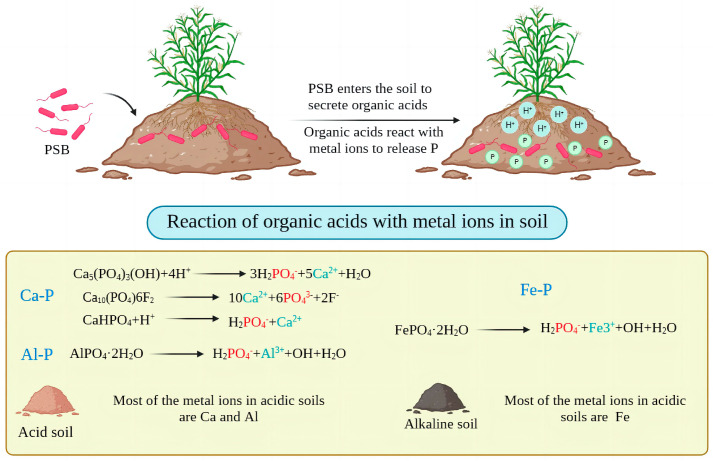
Phosphorus dissolution process in soil.

**Table 5 microorganisms-11-02904-t005:** Secretion of related enzymes by different PSB.

PSB Species	PSB Source	Secreted Enzymes	Reference
*Pantoea* sp.	Forest soil	Phytase	[5]
*Serratia liquiefaciencs*	Laboratory storage	ALP	[93]
*Acinetobacter pittii gp-1*	Laboratory storage	ACP, phytase	[94]
*aryabhattai*	Laboratory storage	ACP	[95]
*Pseudomonas asiatica*	Ant hill soil	ALP	[96]
*Burkholderia* sp.	Zea mays rhizosphere soil	ACP	[97]
*Pseudomonas* sp.	Yellow sandalwood rhizosphere soil	ACP, ALP	[98]
*Enterobacter* sp.	rhizosphere soil of moso bamboo	ACP	[99]

**Table 8 microorganisms-11-02904-t008:** Information on different PSB whole-genome sequencing and enrichment analyses.

PSB	Genome Size (bp)	G+C (%)	Predicted Number of Genes	Acidolysis of Associated Genes	Enzymatic Hydrolysis of Associated Genes	Genes Associated with Phosphorus Cycling	Related Pathways	Reference
*Pseudomonas* sp.	5,617,746	62.86	5097	*GDH*, *ppq*, *gcd*, *gdh*	*ppa*, *PPX*	*PstS*, *PstC*, *PstA*, *PstB*, *PhoU*	Entner–Doudoroff (ED)	[138]
*Serratia marcescens*	5,061,510	59.75	4742	*gcd*, *PQQ*, *ipdC*	*ppx*, *ppa*, *phoA*, *phnX*, *chi*	-	Secreted iron carrier-complete pathway	[139]
*Bacillus subtilis*	4,173,570	43.25	4604	*pyruvate*, *carboxylase*	*phoD*, *phoR*	*pit*, *pstS*, *pstB*	Carbon metabolism, biosynthesis of amino acids	[140]
*Enterobacter cloacae*	4,608,301	54.78	4450	*ppq*, *gcd*, *gdh*	-	*phnG*, *PhnH*, *PhnJ*, *PhnM*, *PhnP*, *PhnI*	-	[141]
*Pseudomonas putida*	5,957,620	61.55	5535	*GDH*, *pqq*, *gcd*, *aceE*, *aceF*, *lpd*, *IDH3*, *idhA*, *dld*, *maeB*	-	*PstS*, *PstC*, *PstA*, *PstB*, *PhoU*	Tricarboxylic acid cycle and pyruvate pathway	[142]
*Pseudomonas fildesensis*	6,789,479	60.82	6028	*GDH*, *pqq*, *gcd*, *aceE*, *aceF*, *lpd*, *IDH3*, *idhA*, *dld*, *maeB*	*phoD*	*PstS*, *PstC*, *PstA*, *PstB*, *PhoU*	Tricarboxylic acid cycle and pyruvate pathway	[142]
*Pantoea*	7,989,160	51.3	4548	*pqq*, *GCD*	-	*phnN*, *phnM*	Indole-3-pyruvate pathway	[143]

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
