# Peer review of "Phosphate-Solubilizing Bacteria: Advances in Their Physiology, Molecular Mechanisms and Microbial Community Effects"

_microorganisms, 2023, doi:10.3390/microorganisms11122904_

Round 1

Reviewer 1 Report

Comments and Suggestions for Authors

The review needs significant improvement. There are a lot of repetitions and unnecessary statements that do not provide interesting information for the reader. The authors should more clearly present studies regarding the molecular mechanisms and genetic basis of phosphorus solubilization. A section on bacterial preparations and their use should be added.

Line 107-108 “PSB are widely distributed and account for 1-60% of phosphorus-solubilizing microorganisms” The meaning of the proposal is not clear. Who are other representatives of phosphate- solubilizing microorganisms if bacteria make up 1-60%? Are these fungi? Why is there such a large range (1-60%)?

Line 110-114 The meaning is not clear.

Line 131, 140 Burkholder? Bakkholderia spp.? Or Burkholderia?

Line 146 “influenza” – What is it? Is this the species name?

Can the term “inter-roots” be replaced by “rhizosphere”? Or are these different concepts?

Line 161-162 What is meant by “isolated from roots”? Are these microorganisms isolated from the rhizosphere, or from the root surface, or these are endophytes?

Line 194-195 “which, in addition to being related to bacterial species, are also regulated by bacterial genes” - The meaning is not clear.

The term “microbial flora”, “microflora” should be avoided.

Line 200-201 The meaning is not clear.

Line 207-208 Needs to be reformulated

Line 240-247 It is necessary to present it more clearly, in short sentences.

Line 265 “neutral phosphatases do not dephosphorylate phosphorus” - The meaning is not clear.

Line 282 Is “iron carriers” means siderophores?

Line 332 and 294 - 2.2.1 - Same headline twice!

Line 360-363 Is it a gene or an enzyme? Most likely, regulation of enzyme activity is described.

Line 369-372, 419-424 It is necessary to reformulate, the meaning is poorly understood, long sentences.

Author Response

Dear Editor,

Thank you very much for giving us the chance to revise our manuscript entitled Phosphate-solubilizing bacteria: advances in their physiology, molecular mechanisms and their microbial community effects” (microorganisms-2710570).

All of the valuable comments provided by reviewers have been taken into full consideration in the revising of the manuscript. All the modifications have been presented by red font. The detailed answers to the comments are summarized separately following this letter. The language of the whole manuscript has also been polished. We hope the revised manuscript could meet the high publication standards of Microorganisms.

Yours sincerely,

Baiyan Cai, PhD

E-mail:[email protected]; Telephone:+86-0451-86608046;

The following is a point-to-point response to the reviewers’ comments on the manuscript entitled Phosphate-solubilizing bacteria: advances in their physiology, molecular mechanisms and their microbial community effects” microorganisms-2710570.

Reviewer #1

Question 1:

The authors should more clearly present studies regarding the molecular mechanisms and genetic basis of phosphorus solubilization. A section on bacterial preparations and their use should be added.

Response:

Thank you for the reviewer’s valuable suggestions. We have described the molecular mechanism of PSB in detail in lines 307-312 of the revised manuscript, and added tables in lines 354 and 403 to illustrate the molecular mechanism more clearly. Added reference to the application of PSB preparations in lines 535 to 559.

Question 2:

Line 107-108 “PSB are widely distributed and account for 1-60% of phosphorus-solubilizing microorganisms” The meaning of the proposal is not clear. Who are other representatives of phosphate- solubilizing microorganisms if bacteria make up 1-60%? Are these fungi? Why is there such a large range (1-60%)?

Response:

Thank you for the reviewer’s valuable suggestions. Relevant literature has been reviewed and new references have been added in lines 114 to 119 of the revised manuscript, the remaining dissolved microbial species have been described and 1-60 per cent has been revised to 1-50 per cent.

Question 3:

Line 110-114 The meaning is not clear.

Response:

Thank you for the reviewer’s valuable suggestions. We have described it more clearly in lines 121-124 of the revised manuscript.

Question 4:

Line 131, 140 Burkholder? Bakkholderia spp.? Or Burkholderia?

Response:

Thank you for the reviewer’s valuable suggestions. Our search of the literature has confirmed that lines 128 and 130 of the revised manuscript read Burkholderia spp. and lines 135 and 142 have been corrected to read Burkholderia .

Question 5:

Line 146 “influenza” – What is it? Is this the species name?

Response:

Thank you for the reviewer’s valuable suggestions. We have checked that "influenza" is incorrect and have corrected it in line 157 of the revised manuscript.

Question 6:

Can the term “inter-roots” be replaced by “rhizosphere”? Or are these different concepts?

Response:

Thank you for the reviewer’s valuable suggestions. After reviewing the literature, we have replaced "inter-roots" with "rhizosphere" throughout the text.

Question 7:

Line 161-162 What is meant by “isolated from roots”? Are these microorganisms isolated from the rhizosphere, or from the root surface, or these are endophytes?

Response:

Thank you for the reviewer’s valuable suggestions. We have replaced "isolated from roots" with "non-rhizosphere soil" in line 173 of the revised manuscript for greater clarity.

Question 8:

Line 194-195 “which, in addition to being related to bacterial species, are also regulated by bacterial genes” - The meaning is not clear.

Response:

Thank you for the reviewer’s valuable suggestions. We have made a change in line 206 of the revised manuscript.

Question 9:

The term “microbial flora”, “microflora” should be avoided.

Response:

Thank you for the reviewer’s valuable suggestions. We have changed "microbial flora" to "microbial community" after reviewing the literature.

Question 10:

Line 200-201 The meaning is not clear.

Response:

Thank you for the reviewer’s valuable suggestions. We have redescribed lines 216 to 220 of the revised manuscript to make their meaning clearer.

Question 11:

Line 207-208 Needs to be reformulated.

Response:

Thank you for the reviewer’s valuable suggestions. We have redescribed lines 216 to 220 of the revised manuscript to make their meaning clearer.

Question 12:

Line 240-247 It is necessary to present it more clearly, in short sentences

Response:

Thank you for the reviewer’s valuable suggestions. We have redescribed the original text in more concise language in lines 253 to 259 of the revised manuscript.

Question 13:

Line 265 “neutral phosphatases do not dephosphorylate phosphorus” - The meaning is not clear.

Response:

Thank you for the reviewer’s valuable suggestions. We have redescribed the original text in lines 276 to 278 of the revised manuscript.

Question 14:

Line 282 Is “iron carriers” means siderophores?

Response:

Thank you for the reviewer’s valuable suggestions. We have replaced "iron carriers" with siderophores throughout the text.

Question 15:

Line 332 and 294 - 2.2.1 - Same headline twice!

Response:

Thank you for the reviewer’s valuable suggestions. We have corrected the title of 2.2.2 in line 356 of the revised manuscript.

Question 16:

Line 360-363 Is it a gene or an enzyme? Most likely, regulation of enzyme activity is described.

Response:

Thank you for the reviewer’s valuable suggestions. We have reviewed the relevant literature, and what is described in this section promotes changes in enzyme activity under the regulatory action of genes.

Question 17:

Line 369-372, 419-424 It is necessary to reformulate, the meaning is poorly understood, long sentences.

Response:

Thank you for the reviewer’s valuable suggestions. We have redescribed the original text in lines 385 to 387 and 431 to 438 of the revised manuscript.

Other changes:

We try our best to improve the manuscript and made some changes in the manuscript. These changes will not influence the content and framework of the paper. And all the modifications have been presented by red font.

We appreciate for Editors/Reviewers’ warm work earnestly, and hope that the correction will meet with approval.

Once again, thank you very much for your comments and suggestions.

In all, I found the reviewers’ comments are quite helpful, and I revised my paper point-by-point. Thank you and the reviewers again for your help.

Reviewer 2 Report

Comments and Suggestions for Authors

The manuscript entitled “Phosphate-solubilizing bacteria: advances in their physiology, molecular mechanisms and their microbial community effects” is a review which aims to summarize the developmental research, species and distribution of phosphorus-solubilizing bacteria, providing an overview of the physiological and molecular mechanisms of phosphorus-solubilizing bacteria and their impact on microecology. To my mind this manuscript is topical and corresponding to the aims and scopes of the Microorganisms journal. This review includes a fairly complete description of the diversity of Phosphate-solubilizing bacteria, the mechanisms of phosphorus accumulation in cells and the ecological role primarily in soil ecosystems based on 142 references.

I believe that after correcting the minor comments below, this review can be published in the journal Microorganisms and will no doubt have a high citation rate.

Here are the comments I found while reading the manuscript.

1. Authors should make the abstract more specific, including the conclusions that were drawn from the review.

2. L110 use superscripts for degree

3. In section 4.3. Give more specific examples of the use of artificial intelligence in this area, the text looks redundant. If there are no examples, then in my opinion it’s better to reduce it to one sentence and not separate it into a separate section

4. The conclusion looks more like perspective. The authors have done a great job of describing current knowledge in the field. It is worth describing this in a few sentences in the conclusion.

Comments on the Quality of English Language

Minor editing of English language required

Author Response

Dear Editor,

Thank you very much for giving us the chance to revise our manuscript entitled Phosphate-solubilizing bacteria: advances in their physiology, molecular mechanisms and their microbial community effects” (microorganisms-2710570).

All of the valuable comments provided by reviewers have been taken into full consideration in the revising of the manuscript. All the modifications have been presented by red font. The detailed answers to the comments are summarized separately following this letter. The language of the whole manuscript has also been polished. We hope the revised manuscript could meet the high publication standards of Microorganisms.

Yours sincerely,

Baiyan Cai, PhD

E-mail:[email protected]; Telephone:+86-0451-86608046;

The following is a point-to-point response to the reviewers’ comments on the manuscript entitled Phosphate-solubilizing bacteria: advances in their physiology, molecular mechanisms and their microbial community effects” (microorganisms-2710570).

Reviewer #2

Question 1:

Authors should make the abstract more specific, including the conclusions that were drawn from the review.

Response:

Thank you for the reviewer’s valuable suggestions. We have added to the summary in lines 16 to 30 of the revised manuscript to make it more detailed.

Question 2:

L110 use superscripts for degree.

Response:

Thank you for the reviewer’s valuable suggestions. We have made the required changes at line 121 of the revised manuscript.

Question 3:

L110 use superscripts for degree. In section 4.3. Give more specific examples of the use of artificial intelligence in this area, the text looks redundant. If there are no examples, then in my opinion it’s better to reduce it to one sentence and not separate it into a separate section.

Response:

Thank you for the reviewer’s valuable suggestions. Since there is not a lot of machine learning related to PSB in the literature at the moment, we have abridged the relevant content in line 555 of the revised manuscript.

Question 4:

The conclusion looks more like perspective. The authors have done a great job of describing current knowledge in the field. It is worth describing this in a few sentences in the conclusion.

Response:

Thank you for the reviewer’s valuable suggestions. We have redescribed the conclusions in 457-464 of the revised manuscript.

Other changes:

We try our best to improve the manuscript and made some changes in the manuscript. These changes will not influence the content and framework of the paper. And all the modifications have been presented by red font.

We appreciate for Editors/Reviewers’ warm work earnestly, and hope that the correction will meet with approval.

Once again, thank you very much for your comments and suggestions.

In all, I found the reviewers’ comments are quite helpful, and I revised my paper point-by-point. Thank you and the reviewers again for your help.